# Uncertainty and Traffic Light Aware Pedestrian Crossing Intention Prediction

## Abstract

Predicting Vulnerable Road User (VRU) crossing intention is one of the major challenges in automated driving. Crossing intention prediction systems trained only on pedestrian features underperform in situations that are most obvious to humans, as the latter take additional context features into consideration. Moreover, such systems tend to be over-confident for out-of-distribution samples, therefore making them less reliable to be used by downstream tasks like sensor fusion and trajectory planning for automated vehicles. In this work, we demonstrate that the results of crossing intention prediction systems can be improved by incorporating traffic light status as an additional input. Further, we make the model robust and interpretable by estimating uncertainty. Experiments on the PIE dataset show that the F1-score improved from 0.77 to 0.82 and above for three different baseline systems when considering traffic-light context. By adding uncertainty, we show increased uncertainty values for out-of-distribution samples, therefore leading to interpretable and reliable predictions of crossing intention.

## 1 Introduction

VRUs are complex participants for an Automated Vehicle (AV) to perceive. The AV should not only be able to detect VRUs, but also understand their underlying intentions and predict their future actions. In addition, several surrounding factors, including incoming traffic and traffic light status, influence VRU behavior. A pedestrian may decide to stop or go at a particular moment based on these conditions. Consider a situation where a pedestrian is standing near the boundary of the curb or walking towards the curb on a traffic light junction looking forward to cross the driving lane of the ego-vehicle. The information about the green traffic light status for the vehicle might help to predict that the pedestrian will keep standing or stop at the curb, i.e., the intention of the pedestrian is not to cross. For this reason, it is necessary to consider surrounding factors like traffic light status in addition to behavioral cues to make an accurate prediction.

Object-based context cues such as pedestrian location over a period of time can provide rich information about VRU motion. But, it is challenging to perceive features like human interactions with the ego-vehicle that can determine its maneuvering. Humans exhibit highly variable motion patterns, and even the same gesture or activity may differ subtly among individuals based on geographic locations. In such a case, it is helpful to divide the task into a smaller sequence of tasks that can be solved independently. In the VRU case, we can learn a model to infer an appearance-invariant representation. The articulated pose of VRUs is one such representation, commonly used in literature for action recognition (Duan et al., 2022), gesture recognition (Mitra & Acharya, 2007), emotion recognition (Shi et al., 2020) and intention prediction (Kotseruba et al., 2021). These object-based features along with surrounding information about the pedestrian can be combined over a temporal domain to generate a reliable predictor for VRU actions in the future. In this paper, we explore this approach and attempt to predict pedestrian crossing intention for a future time horizon of 1-2 s by observing them for a time horizon of 0.5 s.

The handling of VRUs is safety-critical, so it is important to be aware of the uncertainty of the model that predict their behavior as well. Despite much emphasis on safety, deep learning models are often deployed as black box that do not offer reliability and interpretability. As a result, they do not indicate how a system will behave under unknown circumstances. To interpret how the

model behaves in such situations, we intend to predict the uncertainty of each prediction to know the confidence of our model.

## 2 RELATED WORK

### 2.1 ARCHITECTURES FOR PEDESTRIAN CROSSING INTENTION PREDICTION

To build a safer AV system for urban roads, it is important to estimate the crossing intention of a pedestrian, i.e. whether a pedestrian intends to cross/not-cross the road in front of ego-vehicle for a predefined time horizon. Pedestrian crossing intention prediction is mostly treated as a binary task where the goal is to classify between two classes for pedestrian intention, i.e. Crossing (C) or Not Crossing (NC). One of the early works in this direction, as proposed by Rasouli et al. (2017) was to predict crossing action at a given frame using a static representation of the traffic scene and encoding pedestrian looking and walking actions using CNNs. Razali et al. (2021) propose a multitask architecture to estimate the intention and pose of a pedestrian simultaneously using RGB images as input. A multi-task architecture with an encoder-decoder based intention and action prediction to predict pedestrian crossing intent and forecast future behavior of the pedestrians is presented by Yao et al. (2021). The authors also propose an attentive relation network to extract important features from traffic objects and scenes to improve the performance of the intention and action detection framework. Lorenzo et al. (2021b) and Lorenzo et al. (2021a) use vision transformers to encode the non-visual features. They experiment with different types of video encoders and finally fuse the features from the two branches to predict pedestrian crossing intent. Kotseruba et al. (2021) present a pedestrian action prediction model along with a common evaluation criterion. In this paper, the authors evaluate different architectures for pedestrian action recognition, namely static (crossing prediction is made using only last frame in the observation sequence), Recurrent Neural Network (RNN) models, 3D convolution and optical flow based models. They propose a novel architecture based on 3D convolution and multiple RNNs and experiment with different input features like bounding box, local-context, human-pose keypoints and ego-vehicle speed. We base our experiments on this architecture and perform ablation studies to get insights on droupouts and uncertainties. Yang et al. (2022) fuse different phenomena such as sequences of RGB imagery, semantic segmentation masks, and ego-vehicle speed in an optimum way using attention mechanisms and a stack of recurrent neural networks. Achaji et al. (2022) present a framework based on multiple variations of Transformer models to predict the pedestrian street-crossing decision, based on the dynamics of its initiated trajectory, using only bounding boxes as input.

### 2.2 FEATURES USED FOR PREDICTING PEDESTRIAN CROSSING INTENTION

Body language is generally modeled as head orientation, body orientation, posture and gesture which is often used to estimate the pedestrian intention in future. Yang & Ni (2019) use two vision cues to estimate a pedestrian's crossing intention. They propose a looking/not-looking classifier using a 2D convolutional CNN to capture the eye contact between a pedestrian and the ego-vehicle. They also come up with a C/NC classifier based on 3D CNN to model the pedestrian's early crossing action. Roth et al. (2021) propose a method to estimate vehicle-pedestrian path prediction that takes into account the awareness of the driver and the pedestrian towards each other. They extend Dynamic Bayesian Network (DBN) method by Kooij et al. (2014) where they perform path prediction for an individual pedestrian, to the mutual vehicle-pedestrian case. Their results indicate that driver-attention-aware models improve collision risk estimation compared to driver-agnostic models. Human-pose is an intermediate representation which is very useful to determine various human behaviors. Quintero et al. (2017) propose a method to recognize pedestrian intentions such as standing, walking, stopping and starting based on a Hidden Markov Model (HMM). The authors use 3D positions and displacements of 11 skeleton points. They also propose a single frame skeleton estimation algorithm based on point clouds extracted from a stereo pair. Fang & López (2018) use CNN-based pedestrian detection, tracking and pose estimation to predict C/NC action for pedestrians. They use a classifier to predict C/NC using human-pose features. The authors extend their work, to recognize the intention of the cyclists along with the pedestrians (Fang & López, 2019). Mínguez et al. (2018) present a method to predict the future path, pose and intentions of the pedestrians up to a time horizon of 1 s. The authors use balanced Gaussian process dynamic models (BGPDM) to learn 3D time-related information extracted from the skeleton points.

## 2.3 Modeling Uncertainty

To ensure the safety of the AV, it is critical to estimate the uncertainty of model predictions. Gal (2016) show that uncertainty of a deep neural network model, i.e., the epistemic uncertainty or model uncertainty, can be approximated by dropout training. Extending the work of Kendall & Gal (2017), the authors present a Bayesian deep learning framework to estimate both the epistemic and aleatoric uncertainties together. Recently, Djuric et al. (2020) presented a high definition map based approach to predict the future trajectories of the traffic agents, taking the inherent uncertainty of the predictions into account. Peng et al. (2021) propose a method to fine-tune the object detection performance of a deep neural network by calibrating the confidence of the algorithm using model uncertainty. To estimate the model uncertainty, the authors use Monte Carlo Dropout (MCD) method. Model uncertainty can also be used for collecting meaningful data by using active learning as proposed by Kaushal et al. (2019), i.e., high uncertainty samples can be collected and the model can be fine-tuned on these samples to make it more robust through an iterative process.

**Our contributions**:

- We demonstrate that adding traffic light status as additional input improves crossing intention prediction at traffic light junctions by evaluating it on three state-of-the-art baselines (Kotseruba et al. (2021), Yang et al. (2022), Achaji et al. (2022)).

- Introducing uncertainty in crossing intention prediction to produce reliable and interpretable results. This can be used by downstream tasks like sensor fusion and trajectory planning.

- Analyze attention weights distribution and uncertainty for models by sampling different training distributions on PIE dataset.

## 3 Proposed Approach

**Solution Formulation:** The model takes the following observations as input: 2D bounding-box $b_{t-m}, b_{t-m+1}, ...., b_t$ defined by top-left and bottom-right image coordinates, pose of the pedestrian $p_{t-m}, p_{t-m+1}, ...., p_t$, speed of the ego-vehicle $s_{t-m}, s_{t-m+1}, ...., s_t$, local-context $c_{t-m}, c_{t-m+1}, ...., c_t$ where $c$ is the cropped RGB image of the scene around the pedestrian and traffic light status $tl_{t-m}, tl_{t-m+1}, ...., tl_t$ where $t$ is the time-to-event and $m$ is the observation length. Our goal is to predict a crossing action A = {C, NC}, and epistemic uncertainty (U) of the model.

The network architecture of our proposed method as depicted in Figure 1, is inspired by PCPA (Kotseruba et al., 2021). It consists of parallel RNN branches to compute non-visual features such as bounding-box, human-pose and ego-vehicle speed. We add traffic light status as an additional contextual feature. To compute visual features or RGB input which is referred to as local-context, the image crop of the enlarged bounding box (factor 1.5), is processed by 3D convolutions. Each RNN encoder produces a vector $(h_1, h_2, h_3, ..., h_m)$ of hidden states where $h_i = f(x_i, h_{i-1})$.

Attention mechanism is used to combine features from different time frames and modalities as implemented in PCPA (Kotseruba et al., 2021). Temporal attention is used to weigh input for each time step and focus on important temporal events. It is learned for $m = 16$ hidden temporal states for each of the RNN branches. The attention mechanism applied here is the Luong's multiplicative attention (Luong et al., 2015). The goal is to get an attention weight vector $\alpha$ with length equal to observation length $m$. Further, modality attention is learned for weighing all the different input modalities in a similar manner. Here, the attention mechanism is applied to all the final vectors from 3D convolution branch and temporal attention outputs i.e. human-pose, bounding-box, ego-vehicle speed and traffic light status.

Epistemic or model uncertainty which denotes under-represention of samples in training distribution, is computed during inference by running a Monte Carlo sampling over the network for $N$ number of times (Gal, 2016).

$$P = p(y = Y|x, D_{train}) \tag{1}$$

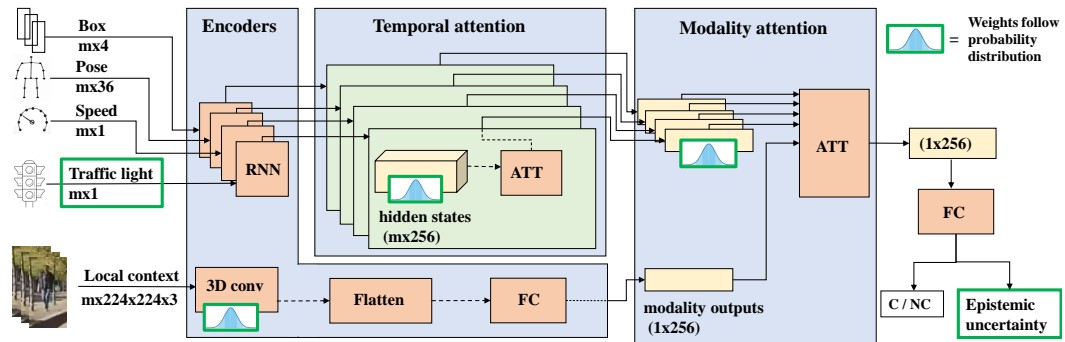

Figure 1: Architecture diagram of the proposed approach inspired by PCPA (Kotseruba et al., 2021). RNN: Recurrent Neural Network, FC: Fully Connected Layer, ATT: Attention block. Our contributions are highlighted in green borders. We incorporate the traffic light status as an additional input. Dropouts at temporal and modality attention blocks are added which are also active during run-time for performing Monte Carlo simulation to give probabilistic output for each simulation run.

$$H[y|x, D_{train}] = -\sum_Y P \log P \tag{2}$$

where $H$ represents the predictive entropy and is the measure of model uncertainty, $Y$ represents the output classes C/NC and $P$ represents the final probability vector for C and NC class which is then mapped to crossing action A = {C, NC} for the given observation horizon. It is obtained by averaging all output probability vectors from $N$ forward passes.

## 4 EXPERIMENTS

### 4.1 DATASET

In this work, we use Pedestrian Intention Estimation (PIE) by Rasouli et al. (2019) which is a large public benchmark dataset for studying pedestrian behavior. It contains 6 hours of continuous footages recorded in Canada under clear weather conditions. It provides annotations for all pedestrians sufficiently close to the road who may or may not attempt to cross in front of the ego-vehicle. We follow the data split defined by Rasouli et al. (2019), i.e., videos from set01, set02 and set04 are used for training (TRAIN split), set05 and set06 for validation (VAL split) and set03 for testing (TEST split). The number of pedestrian tracks in PIE is 880, 243 and 719 in TRAIN, VAL and TEST splits, respectively.

We use following explicit features that are annotated in PIE dataset: bounding-box coordinates, ego-vehicle speed, and attributes for the scene e.g. local-context and traffic light. In addition, human-pose information here is generated by OpenPose (Cao et al., 2017). Thus, we have 18 skeleton point coordinates which are concatenated into a 36D feature vector. PIE dataset has annotations for

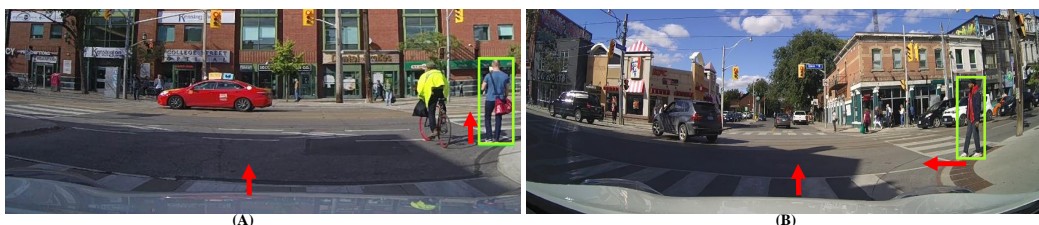

Figure 2: Pedestrian crossing parallel to the direction of motion of ego-vehicle (**A**) and perpendicular to the direction of motion of ego-vehicle (**B**). The arrows in red denote the direction of motion of pedestrian and ego-vehicle.

traffic light as an entity, with traffic light status being undefined, red, yellow or green. We associate the traffic light status to the corresponding pedestrians in that frame using frame-ids. This traffic light status was mapped to the relevant pedestrians using a semi-automatic labeling technique as follows: We first assign body-orientations for all pedestrians in the PIE dataset. This was achieved by adding a body-orientation predictor to the VRU Pose-SSD (Kumar et al., 2021) model which was trained on body-orientation labels in TDUP dataset (Wang et al., 2021). Based on the values of body-orientation, we label the pedestrians to be crossing parallel or perpendicular to the direction of motion of ego-vehicle as shown in Figure 2. We then associate the traffic light status in the scene, for perpendicular crossings and not-crossing pedestrians, as the traffic light in the scene is valid only for these pedestrians. This is because the parallel crossings are not regulated by the front traffic lights which is visible to the ego-vehicle. For pedestrians crossing parallel to the direction of motion of the ego-vehicle, we encode the traffic light status to be undefined during training and inference.

## 4.2 COMPARISION OF RESULTS ON DIFFERENT ARCHITECTURES

**Metrics:** We report the results using standard binary classification metrics: accuracy, Area Under Curve (AUC) and F1-score ($F1$) defined as $F1 = \frac{2 \times (precision \times recall)}{(precision + recall)}$.

To evaluate crossing predictions when adding traffic light status to inputs, we experiment with three different architectures: PCPA (Kotseruba et al., 2021), Feature-Fusion (Yang et al., 2022) and TED model (Achaji et al., 2022). The process of dataset preparation remains same across these architectures. We report consistent improvement across all metrics when incorporating traffic light status as additional input. There is a improvement of 5% (**CIPTLU$_{mt}$**, $F1 = 0.82$), 6% (**Feature Fusion**, $F1 = 0.83$) and 6% (**TED**, $F1 = 0.83$) in the F1-score respectively as shown in Table 1. Since, the dataset is not balanced, i.e., there are more not-crossing pedestrians as compared to crossing pedestrians (1322 NC vs. 512 C), we use F1-score for performance comparision of the models. We strongly believe that adding traffic light status to any architecture will improve its performance, as the model learns better representations to predict pedestrian behavior for stop-and-go transitions at traffic light junctions. Although, we compare these three architectures, we now focus our experiments and analysis of attention-weights distribution and uncertainty only on PCPA as it allows us

Table 1: Accuracy (Acc), Area Under Curve (AUC), and F1-score comparison with baseline models (1., 7. and 9.). All metrics reported here are on the TEST set of PIE dataset which is described under the *dataset* sub-section in Section: 4. TL status refers to presence or absence of traffic light status in the inputs and MC Dropout refers to adding Monte Carlo Dropouts to the model. Here, *modal / temp* signifies dropout layers are applied inside the modality attention block or temporal attention block respectively as shown in Figure 1. We extend PCPA to CIPU: Crossing Intention Prediction with Uncertainty, CIPTL: Crossing Intention Prediction with Traffic Light, CIPTLU: Crossing Intention Prediction with Traffic Light and Uncertainty. The subscripts $m$ and $t$ denotes the blocks (*modal / temp*) where the dropout is applied. The models in boldface represent the best setting for that architecture. Inputs to Feature Fusion are same as that of PCPA wheres inputs to TED constitutes only bounding-boxes other than traffic light status.

| S.No. | Model | TL status | MC Dropout | Acc | AUC | F1 |
|---|---|---|---|---|---|---|
| 1. | PCPA (Kotseruba et al., 2021) | no | no | 0.87 | 0.86 | 0.77 |
| 2. | CIPTL | yes | no | 0.89 | 0.86 | 0.81 |
| 3. | CIPU$_m$ | no | modal | 0.86 | 0.85 | 0.77 |
| 4. | CIPTLU$_m$ | yes | modal | 0.89 | **0.87** | 0.81 |
| 5. | CIPTLU$_t$ | yes | temp | **0.90** | **0.87** | 0.81 |
| 6. | **CIPTLU$_{mt}$** | yes | modal, temp | **0.90** | **0.87** | **0.82** |
| 7. | Feature Fusion (Yang et al., 2022) | no | no | 0.87 | 0.84 | 0.78 |
| 8. | **Feature Fusion** | yes | no | **0.91** | **0.87** | **0.83** |
| 9. | TED (Achaji et al., 2022) | no | no | 0.86 | 0.86 | 0.78 |
| 10. | **TED** | yes | no | **0.89** | **0.89** | **0.83** |

to experiment with several inputs and also uses attention mechanism which gives deeper insights on representations learned by the model.

### 4.3 EXPERIMENTS ON PCPA

**Inputs, model architecture and training**: By adding traffic light status as input to PCPA and adding MCD on temporal and modality attention blocks, we report a improvement of 5% ($F1 = 0.82$) in the F1-score as shown in Table 1 (**CIPTLU$_{mt}$**). Following Kotseruba et al. (2021), we use RNN based model with 256 hidden units for each non-visual input and a conv3D network for encoding RGB input. The number of observation frames ($m$) is set to 16 ($\approx 0.5\,\text{s}$) and the time-to-event is $30 - 60$ frames (1-2 s). Dropout of $0.5$ is added before temporal and modality attention. L2 regularization of $0.001$ is added to the final dense layer. The batch-size is set to $8$ and model is trained with Adam optimizer. It is trained for 60 epochs with $5 \times 10^{-5}$ learning rate.

**Effect of Monte Carlo Dropout:** As mentioned in Section 3, Monte Carlo sampling is performed $N$ number of times during inference where $N = 10$. Even for higher values of $N$, we observe that the distribution of uncertainty remains same. We experiment with dropout layer at several places like inside the temporal attention block and modality attention block as shown in the Table 1. Applying dropout for both temporal and modality attention gives the best results ($F1 = 0.82$) for PCPA model. Further, we add dropout before the last FC layer but it neither improves nor degrades the performance ($F1 = 0.81$). We also perform experiments with different values of dropouts, namely $0.25$, $0.50$ and $0.75$ where we observe that dropout value of $0.50$ gives the best results whereas the F1-score either degrades or remains constant for other values.

**Ablation study on model uncertainty:** Results of experiments to analyze the varation of samples with high uncertainty values w.r.t. time horizon are shown in Figure 3. Since pedestrian behavior becomes more complex when predicting for longer horizons, the uncertainty increases as the time horizon increases.

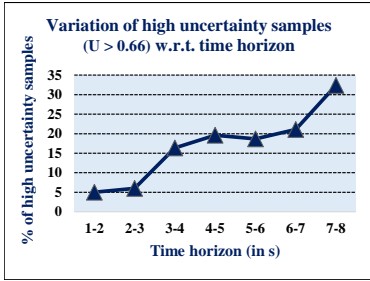

Figure 3: Percentage of samples with high uncertainty increases as the training and prediction horizon boundary is increased.

We perform experiments to analyze the effect of under-represented samples on uncertainty. In-order to achieve this, we create modified TRAIN splits by removing a type of distribution from the PIE TRAIN split. We perform following three experiments by creating under-representations with-respect-to bounding-box, ego-vehicle speed and traffic light status: For bounding-box (Figure 4: A), we remove all the samples from the TRAIN split with pedestrian height less than 100 pixels to get the modified TRAIN split for bounding-box experiment. We use this modified TRAIN split to train a new model. Uncertainties are then filtered for the samples with pedestrian height less than 100 pixels in the TEST split. Similarly, we perform experiments by removing all samples with speed equal to zero (Figure 4: B) and removing all samples with green-light status in the TRAIN split (Figure 4: C). When we compare the uncertainties on same samples from the original model, we observe that the uncertainties are clearly shifted towards larger values for the under-represented samples. This demonstrates the effectiveness of model uncertainty in estimating out-of-distribution samples in the training distribution.

**Model uncertainty and misclassification rate:** We observe that increase in uncertainty has a strong correlation with misclassification rate (ratio of number of incorrect predictions to total number of predictions) for crossing action prediction. This remains true for both the models with and without traffic light status (see Figure 5). This property of uncertainty can be utilized for ignoring samples

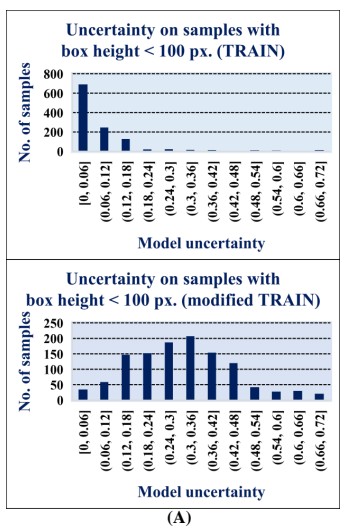 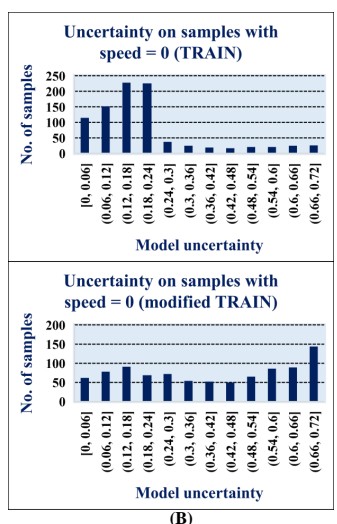 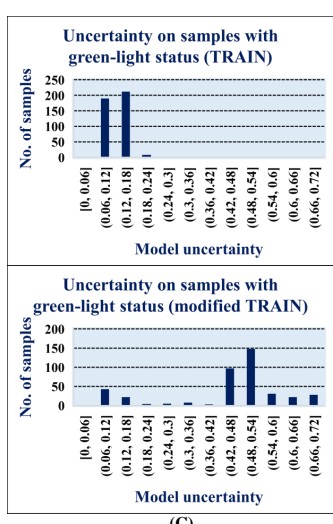

Figure 4: Ablation study on uncertainty with inputs. The output uncertainties are from the model **CIPTLU_mt** trained on datasets mentioned in the figure, i.e., **TRAIN** and **modified TRAIN** which represents model trained on TRAIN split and reduced TRAIN splits respectively. The samples represented in the graphs are all the samples with mentioned settings on box height, speed and traffic light status from the TEST split. **(A)**, **(B)** and **(C)** correspond to uncertainty ablation experiments with bounding-box, ego-vehicle speed and traffic light status respectively.

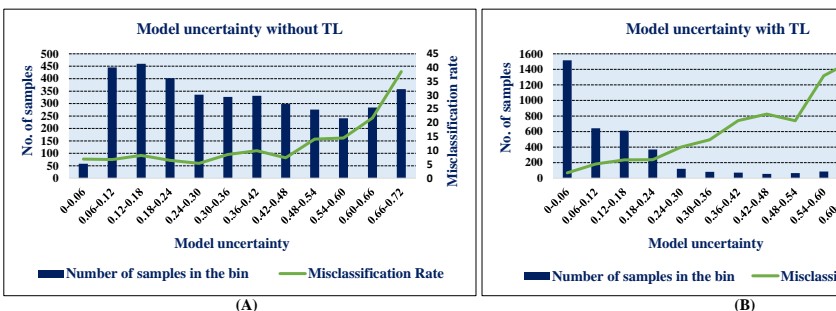

Figure 5: Misclassification rate of model uncertainty without TL status (**A**) and with TL status (**B**). The misclassification rate increases with model uncertainty for both models. The uncertainty values recorded in model with traffic light shift towards lesser values, which shows the model is more certain when using traffic light status.

with high uncertainty and spawning crossing action for pedestrians only with reasonable uncertainty for downstream tasks like sensor fusion and trajectory planning.

**Analysis of attention weights:** We compare the attention weights for the inputs of the model and found that local-context features still dominate for few samples in the model trained with traffic light status. There has not been much change in the attention weights of the earlier inputs like local-context, bounding box, human-pose and ego-vehicle speed except that they are pushed towards lower values to incorporate the weight given to traffic light status. But most importantly, there are no cases where traffic light status dominates as a feature as shown in Figure 6. This distribution is important, as it shows that adding traffic light status as input does not shift the weights entirely towards traffic light. Domination of traffic light in attention weight distribution would have meant that the model ignores all other cues which might be crucial in scenarios where the pedestrians are violating the traffic light rules. We observe that PIE dataset does not contain cases where the pedestrian violates traffic light. Such scenarios play an important role in modelling a rare pedestrian behavior (jaywalking on signalized junctions) which can lead to accidents.

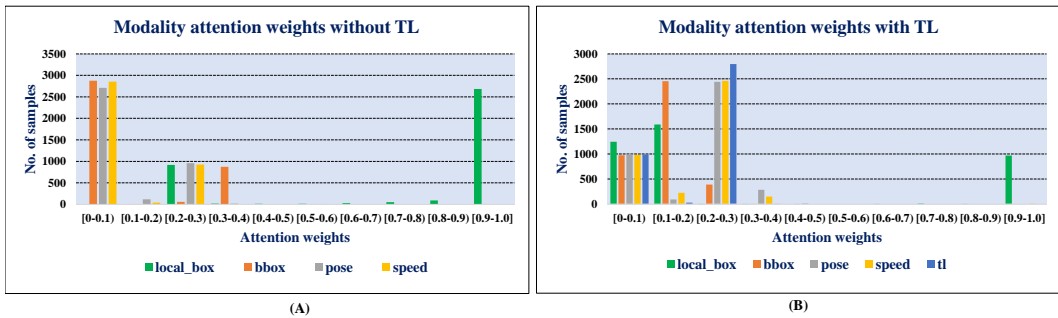

Figure 6: Attention weights distribution in model trained without (**A**) and with (**B**) traffic light status. Local-context still dominates the weights distribution for some samples. Additionally traffic light status does not dominate the weights distribution in any of the samples.

**Qualitative results:** Some examples of low and high uncertainties are shown in the Figure 7. Many low uncertainty samples belong to the category of most common scenarios like NC scenarios where the pedestrian is standing near the curb (Figure 7: A) and the car is at moderate speed (greater than 30 kmph). It is observed that uncertainty is high for samples with shorter pedestrian height (children) (Figure 7: B), occlusion (Figure 7: C) and car taking a turn (Figure 7: D) as these category of samples are under-represented in PIE dataset.

A comparison of success and failure cases on PIE TEST split for the models trained without and with traffic light status (left and right snippets repectively) is shown in the Figure 8. The success cases occur when the traffic light status is either green or red (Figure 8: A and B). On the other hand, failure cases are samples with yellow light status (Figure 8: C). This is because the PIE dataset contains low amount of samples with yellow light as compared to red and green light (as yellow light is a transition light between red and green). The predictions become more stable across the $0.5\,\text{s}$ prediction window (Figure 8: D) which is attributed to improvement in F1-score. Also, the uncertainties for samples with traffic light (only for red and green light, as yellow light is under-represented in PIE dataset) have drastically reduced for model trained with traffic light status (compare U values for samples in left and right snippet). This shows the confidence of the model has increased when adding traffic light status. This is also evident for the model uncertainty graphs shown in Figure 5, as the overall uncertainty distribution shifts towards lower values in the model trained with traffic light status.

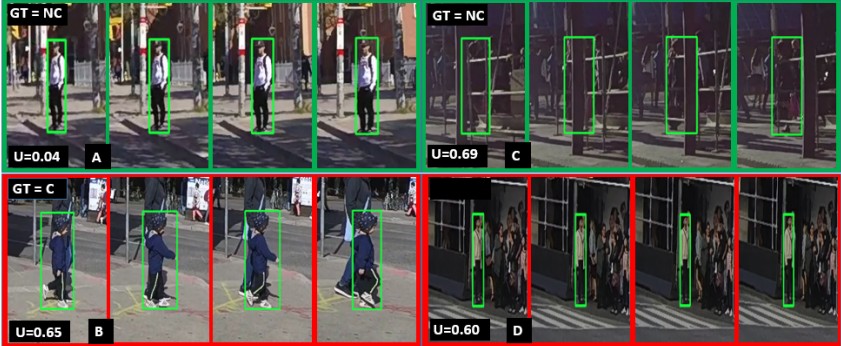

Figure 7: Examples for low (**A**) and high uncertainty (**B**), (**C**), (**D**) samples from PIE dataset. GT: Ground Truth (C/NC) and epistemic uncertainty (U) are marked on top and bottom of the snippets respectively. Snippets encapsulated in red indicates wrong model prediction and green encapsultation indicates correct model prediction.

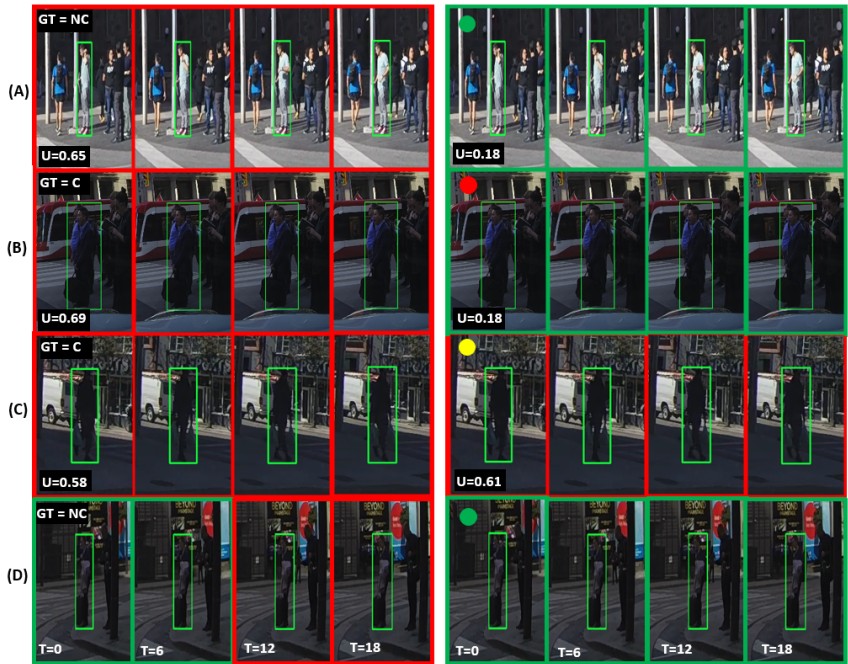

Figure 8: Comparision of predictions from the model without traffic light status (**CIPUm**: left) and with traffic light status as input (**CIPTLUmt**: right). Refer to Figure 7 caption for explanation to border color and abbreviations. Snippets with colored circle on top (green/ red/ yellow) indicates the traffic light status for that sample. Snippets in (**A**), (**B**) and (**C**) belong to the same subsample of the observation window (16 frames $\approx 0.5$ s) whereas in (**D**), the snippets are from subsequent subsamples with an overlap of 10 frames ($\approx 0.3$ s).

## 5  CONCLUSION AND FUTURE WORK

We demonstrate that adding traffic light status as input improves crossing intention predictions as the model learns representations for traffic light status as an additional contextual feature that influence pedestrian behavior. We prove this by showing improvement when incorporating traffic light status across three baseline models for crossing prediction. We also demonstrate that a model becomes more interpretable and robust when it predicts uncertainty for the crossing intention prediction, as it provides a way to assess the reliability of the predictions by estimating a factor by which the sample is represented in the model. Further, we demonstrate that traffic light status does not dominate completely over other inputs by analyzing the attention weights distribution for the inputs. We also show that uncertainty of pedestrian intention prediction is directly proportional to the length of prediction horizon.

We also observe that the PIE dataset does not cover a variety of traffic light scenarios. As pedestrians are likely to break rules on the road, one of the most important scenario that is missing in the dataset is the jaywalking cases at traffic light junctions. With uncertainty, our proposed model can deal with jaywalking to a limited degree. We argue, that preparing the model explicitly for these scenarios can ensure more accurate predictions at traffic light junctions. Therefore, there is a need for a dataset that has sufficient samples for jaywalking with and without traffic light junctions. We aim to prepare such dataset as one of our future works. Further, we want to optimize the model to avoid repetitive computations at runtime to perform MC sampling for uncertainty predictions. This work provides us with the necessary framework and evidence to further research uncertainty estimations and modelling surounding factors like traffic light status for pedestrian crossing intention prediction.

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
