# OpenReview forum: "Uncertainty and Traffic Light Aware Pedestrian Crossing Intention Prediction"
_ICLR.cc/2023/Conference — Submitted to ICLR 2023_

### Official Review · Reviewer_DUfQ · 2022-10-13

**Confidence:** 4
**Correctness:** 3
**Technical Novelty And Significance:** 2
**Empirical Novelty And Significance:** 2
**Recommendation:** 5

**Clarity, Quality, Novelty And Reproducibility:**

The novelty of this paper is rather low. The network architecture proposed in this paper is almost exactly the same as the PCPA model in Kotseruba et al. The only difference is the addition of the traffic light inputs, which is a trivial change.

**Strength And Weaknesses:**

--- Strengths

- Pedestrian crossing intention prediction is an important problem. This paper presents a model that improves over the state-of-the-art baseline.


--- Weakness

- Novelty is the biggest issue of this paper. The network architecture proposed in this paper is almost exactly the same as the PCPA model in Kotseruba et al. The only difference is the addition of the traffic light inputs, which is a trivial change.

- The improvement over the baseline is relative minor.

- Although the evaluation result shows the proposed model improves the accuracy of pedestrian crossing prediction, but I doubt the real impact of this work. When the light on the ego lane is red, the ego vehicle will need to stop anyway, regardless of whether the pedestrian will cross the street or not. With the additional traffic light inputs, the model will unsurprisingly become better at predicting the crossing intention for those pedestrians crossing perpendicular to the ego vehicle, but the predictions for those pedestrians rarely matter. What truly matters is to predict whether a pedestrian will cross the street when the ego vehicle is moving (e.g., jaywalking).

- The proposed model predicts the epistemic uncertainty, which can be addressed by adding more training data. What matters more is the aleatoric uncertainty.

- The PIE dataset has only 880 training samples, which is way too small for this task. With a larger dataset, epistemic uncertainty will not matter that much anymore.

**Summary Of The Paper:**

This paper presents a method for predicting pedestrian crossing intentions. The model is an extension of the PCPA model proposed in Kotseruba et al. On top of the PCPA model, they 1) added the traffic light inputs and 2) used the dropout method to predict the epistemic uncertainty.

The authors evaluated the proposed model on the PIE dataset, and the model is shown to improve the prediction accuracy over the PCPA baseline. Moreover, the predicted epistemic uncertainties are shown to correlate with the misclassification rate.

**Summary Of The Review:**

The novelty of this paper is rather low, so I am giving it a weak reject.

---

### Official Review · Reviewer_MM9u · 2022-10-25

**Confidence:** 5
**Clarity, Quality, Novelty And Reproducibility:** Please see the detailed comments in t…
**Correctness:** 3
**Technical Novelty And Significance:** 1
**Empirical Novelty And Significance:** 2
**Recommendation:** 3

**Strength And Weaknesses:**

Strengths:
- Relevant and timely work
- Promising results using the new TL data

Weaknesses:
- Very limited novelty
- The paper can be written better, it is not fully self-contained

**Summary Of The Paper:**

The authors explore the benefits of using traffic light (TL) information for predicting the probability of a pedestrian crossing the street. They use existing methods and architectures to which the TL inputs are added, and show the benefits of this new information. In addition, they explore the epistemic uncertainty of their model, and correlate it with the model performance (such as misclassification rate).

**Summary Of The Review:**

The work has very limited novelty. The authors use existing methods (such as the architectures that they consider), and apply a new TL input without any deeper considerations or insights. In addition, they explore using dropout to estimate epistemic uncertainty which is a well-known method and no novelty can be claimed there. In fact, the authors only focus on epistemic and disregard aleatoric uncertainty, which would help and add some more novelty to the work.
Detailed comments follow:
- The authors say "even the same gesture or activity may differ subtly" to say that new features are needed to capture interactions, yet a few sentences later they say that this motivates them to use pose features which doesn't really help in that regard.
- The architecture that is used is not well explained, and the authors should make their work more self-contained. E.g., they don't clarify well how are attentions implemented and what attends over what. They do provide a reference, however given already low novelty and the fact that this attention is explored quite a lot in the experiments, this important part of their methodology should be better explained in the text.
- In relation to what are equations (1) and (2) added? It is unclear, they seem to be suddenly added to the text.
- Traffic light info and its relevance should be discussed much more, given that that is the main topic of the work.
- As explained at the end of Section 4.1, a lot of TL data is actually ignored (when it comes to pedestrians walking in parallel). More statistics about the data should be added to better quantify and understand this aspect.
- How is MCD actually used in the work? Do you run multiple times and then average? This is not explained.
- The authors point out that rare appearance of jaywalkers in the used data is affecting their results, however they do not dig deeper into this aspect which would be an interesting addition to the work. They leave it for future work which is unfortunate.
- In general, the experimental results are not adding a lot of value, as many are already well known and discussed before (such as a large part of uncertainty results). I did find Fig 4 interesting however.

---

### Official Review · Reviewer_ZTh8 · 2022-10-28

**Confidence:** 3
**Clarity, Quality, Novelty And Reproducibility:** The writing is clear, the novelty is …
**Correctness:** 3
**Technical Novelty And Significance:** 2
**Empirical Novelty And Significance:** 2
**Recommendation:** 3

**Strength And Weaknesses:**

The problem of Vulnerable Road User is not common in literature.
The sole contribution of the work is in introducing the usage of traffic light status through attention into previous models that is used to solve this problem.

The model was evaluated on a single dataset ignoring other datasets like JAAD[1].
The introduced model is the same as used in [1] with exception to adding the traffic status as the work indicates.
Incremental performance where AUC is increased by 1-3% same for the accuracy. There is no std reported for the results to check if these numbers are due to different random seeds or not.
The analysis of method and different components section 4.3 is quite good.

[1] "Benchmark for Evaluating Pedestrian Action Prediction"

**Summary Of The Paper:**

The work introduced the usage of traffic signals into the problem of predicting the crossing intention. The work has a good amount of ablation study.

**Summary Of The Review:**

The work need more results on other datasets.
There is no significant model design introduced.

---

### Decision · Program_Chairs · 2023-01-20

**Decision:**

Reject

**Justification For Why Not Higher Score:**

Based on the recommendation of the reviewers and the absence of author feedback, all reviewers agreed that the paper tackles an important problem but there are many weaknesses in order to accept the paper.

**Justification For Why Not Lower Score:**

N/A

**Metareview: Summary, Strengths And Weaknesses:**

# Summary
The proposed method presents a method for predicting pedestrian crossing intentions by exploring the benefits of using traffic light information. The model is an extension of the PCPA model proposed in Kotseruba et al. On top of the PCPA model, they 1) added the traffic light inputs and 2) used the dropout method to predict the epistemic uncertainty. The proposed model is evaluated on the PIE dataset, and the model is shown to improve the prediction accuracy over the PCPA baseline. Moreover, the predicted epistemic uncertainties are shown to correlate with the misclassification rate.
# Strengths:
- Pedestrian crossing intention prediction is an important and relevant problem.
- Promising results using the new TL data
-This paper presents a model that improves over the state-of-the-art baseline.
# Weaknesses:
- Very limited novelty. The network architecture proposed in this paper is almost exactly the same as the PCPA model in Kotseruba et al. The only difference is the addition of the traffic light inputs, which is a trivial change.
- The improvement over the baseline is relative minor.
- The proposed model predicts the epistemic uncertainty, which can be addressed by adding more training data. What matters more is the aleatoric uncertainty.
- The PIE dataset has only 880 training samples, which is way too small for this task. With a larger dataset, epistemic uncertainty will not matter that much anymore.